# Spin blockade and phonon bottleneck for hot electron relaxation observed in *n*-doped colloidal quantum dots

Junhui Wang [1✉], Lifeng Wang[1,2], Shuwen Yu[1], Tao Ding [1,3], Dongmei Xiang[1] & Kaifeng Wu [1✉]

Understanding and manipulating hot electron dynamics in semiconductors may enable disruptive energy conversion schemes. Hot electrons in bulk semiconductors usually relax via electron-phonon scattering on a sub-picosecond timescale. Quantum-confined semiconductors such as quantum dots offer a unique platform to prolong hot electron lifetime through their size-tunable electronic structures. Here, we study hot electron relaxation in electron-doped (*n*-doped) colloidal CdSe quantum dots. For lightly-doped dots we observe a slow $1P_e$ hot electron relaxation (~10 picosecond) resulting from a Pauli spin blockade of the preoccupying $1S_e$ electron. For heavily-doped dots, a large number of electrons residing in the surface states introduce picosecond Auger recombination which annihilates the valance band hole, allowing us to observe 300-picosecond-long hot electrons as a manifestation of a phonon bottleneck effect. This brings the hot electron energy loss rate to a level of sub-meV per picosecond from a usual level of 1 eV per picosecond. These results offer exciting opportunities of hot electron harvesting by exploiting carrier-carrier, carrier-phonon and spin-spin interactions in doped quantum dots.

[1] State Key Laboratory of Molecular Reaction Dynamics and Dynamics Research Center for Energy and Environmental Materials, Dalian Institute of Chemical Physics, Chinese Academy of Sciences, Dalian 116023 Liaoning, China. [2] University of the Chinese Academy of Sciences, Beijing 100049, China. [3] National Synchrotron Radiation Laboratory, University of Science and Technology of China, Hefei 230029 Anhui, China. ✉email: wjh@dicp.ac.cn; kwu@dicp.ac.cn

Hot electrons carry large kinetic energies compared to band edge electrons. They have been a fascinating subject of research for solid state materials because of their potentials in disruptive energy conversion technologies. For example, solar cells utilizing hot electrons can break the Shockley-Queisser efficiency limit of conventional solar cells[1–3]. Exciting opportunities are also associated with hot electron photocatalysis where highly-energized electrons in semiconductor or metal nanoparticles can enable various useful chemical transformations[4–7]. However, hot electrons usually relax on a sub-ps timescale due to electron-electron, electron-phonon and/or electron-impurity scatterings, making it highly challenging to harvest hot electrons. Therefore, understanding and manipulating hot electron dynamics is essential for hot-electron-related applications. Recent studies suggest that long-lived hot carriers might be attainable in lead halide perovskites via a unique polaronic mechanism[8–10], but in most cases a hot phonon bottleneck available at high excitation densities is required to prolong the hot carrier lifetime[11–17].

Historically, semiconductor nanocrystals or quantum dots (QDs) have been extensively studied under the context of hot electron relaxation and transfer[18–24]. This is because quantum confinement effect in QDs leads to discrete energy levels with inter-level spacing reaching hundreds of meV[25–27]. As a result, hot electron relaxation requires emission of many phonons, which is inefficient and is called a phonon bottleneck (Fig. 1a)[20,22]. However, hot electrons in various types of QDs still relax on a sub-ps timescale, because strong confinement opens up additional ultrafast relaxation channels[28–34]. For prototypical CdSe colloidal QDs, it has been suggested that a hot electron can lose its excessive energy by transferring it to a hole, and the latter can rapidly relax via phonon emission because of a large density of states in the valence band (Fig. 1a)[28,35,36]. Along this line, in our recent work, we have demonstrated that in copper-doped CdSe QDs, hole capturing by the copper dopants can effectively compete with the sub-ps electron-to-hole energy transfer, resulting in hot electron lifetime as long as 8.6 ps[37]. However, a large portion of hot electrons still relax within a few ps, suggesting that copper-localized holes could still accept energy from hot electrons. This motives us to further search and/or design QD systems for long-lived hot electrons.

Carrier-doped QDs represent a unique platform to realize peculiar optical and electronic properties by exploiting the interaction between preoccupying band edge carriers and injected carriers/excitons. For example, electron-doped (also called $n$-doped) QDs have enabled the concept of "zero-threshold optical gain" because the preexisting electrons can, in principle, completely block band edge absorption, on the basis of which any weak photoexcitation immediately induces an optical gain[38–41]. We realize that a similar "blockade" idea can be applied to hot electron relaxation. With one electron preoccupying the $1S_e$ level, the photogenerated $1P_e$ hot electron has 50% likelihood to possess the same spin as the preoccupying electron; in this case, there is a Pauli spin blockade for hot electron relaxation and the electron spin has to be flipped before it could relax by electron-to-hole energy transfer (Fig. 1b, left). With two doped electrons fully occupying the $1S_e$ level, the $1P_e$ hot electron relaxation channel is completely blocked by state-filling (Coulomb blockade) and it would not relax until multi-carrier Auger recombination annihilates one $1S_e$ electron (Fig. 1b, middle). Alternatively, a doped $1S_e$ electron can be directly excited to the $1P_e$ level using a mid-infrared (mid-IR) photon; the resulting $1P_e$ electron is fully decoupled from any electron or hole and it should be very long-lived because of a phonon bottleneck (Fig. 1b, right). Thus, there are plenty of opportunities to engineer hot electron dynamics by exploiting carrier-carrier, carrier-phonon and spin-spin interactions in carrier-doped QDs.

Here we report a time-resolved spectroscopy study of hot electron relaxation dynamics in photochemically $n$-doped colloidal CdSe QDs. For lightly-doped QDs we observe a slow $1P_e$ hot electron relaxation (~10 ps) consistent with the expected Pauli spin blockade. For heavily-doped QDs, a large number of electrons residing in the surface states induce ps Auger recombination that annihilates the valence band hole. This allows us to observe 300-ps-long $1P_e$ hot electrons, which we attribute to the phonon bottleneck effect, representing three orders of magnitude lengthening as compared to neutral QDs. The hot electron energy loss rate is brought to a level of sub-meV/ps from a usual level of ~1 eV/ps.

## Results and discussion

**Characterizations of neutral and $n$-doped QDs.** Zinc blende CdSe QDs of varying sizes were synthesized using a literature method;[42] see Methods for details. Transmission electron microscope (TEM) images indicate that the diameters are tuned from 4.2 to 5.5 nm (Supplementary Fig. S1). Figure 2a shows the absorption spectrum of the 5.5-nm QD. By taking a second derivative of the spectrum, we reveal at least four absorption peaks that can be assigned to the $1S_e$-$1S_{3/2,h}$, $1S_e$-$2S_{3/2,h}$, $1P_e$-$1P_{3/2,h}$, and $1S_e$-$3S_{1/2,h}$ excitonic transitions in the order of increasing energy[43,44]. Absorption spectra of other size QDs are shown in Supplementary Fig. S2.

QDs were negatively doped using a photochemical approach[45]. Under continuous illumination, the photogenerated hole in the QD is scavenged by a sacrificial donor, triethylborohydride (LiEt₃BH), leaving behind the electron in the QD; see "Methods". The number of doped electrons can be controlled by the amount of LiEt₃BH added. Figure 2b shows the absorption spectra of two $n$-doped 5.5-nm QD samples along with that of the pristine one. Bleaching of the $1S_e$-$1S_{3/2,h}$ and $1S_e$-$2S_{3/2,h}$ absorption peaks is an indication of successful electron injection into the $1S_e$ level. Using the bleaching amplitudes and accounting for a two-fold degeneracy of the $1S_e$ level[45], the average numbers of doped $1S_e$ electrons are calculated to be 0.5 and 1.3 for samples $n$-doped-1 and $n$-doped-2, respectively. Importantly, a previous "electron-titration" experiment shows that in addition to band-edge

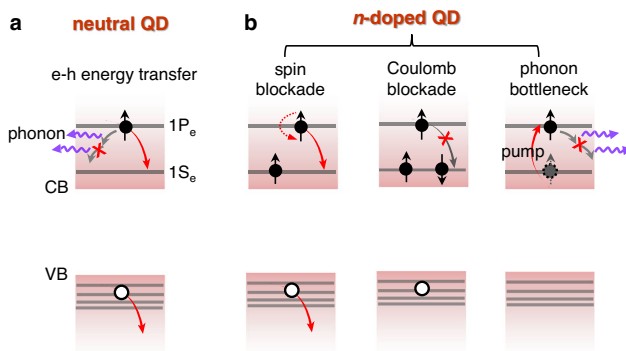

**Fig. 1 Slow hot electron relaxation in $n$-doped QDs. a** In a typical CdSe QD, an $1P_e$ hot electron relaxes on a sub-ps timescale through electron-to-hole energy transfer, breaking the otherwise expected phonon bottleneck. **b** In $n$-doped QDs, slow hot electron relaxation is expected due to the preoccupying $1S_e$ electrons. For a QD doped with one $1S_e$ electron (left), there is a 50% probability that the $1P_e$ electron would relax slowly due to a Pauli spin blockade; for a QD doped with two $1S_e$ electrons (middle), relaxation of the $1P_e$ electron is completely blocked due to state-filling; alternatively, a preoccupying $1S_e$ electron can be excited into the $1P_e$ level (right), which would be followed by slow relaxation because the phonon bottleneck cannot be bypassed for lack of a valence band hole.

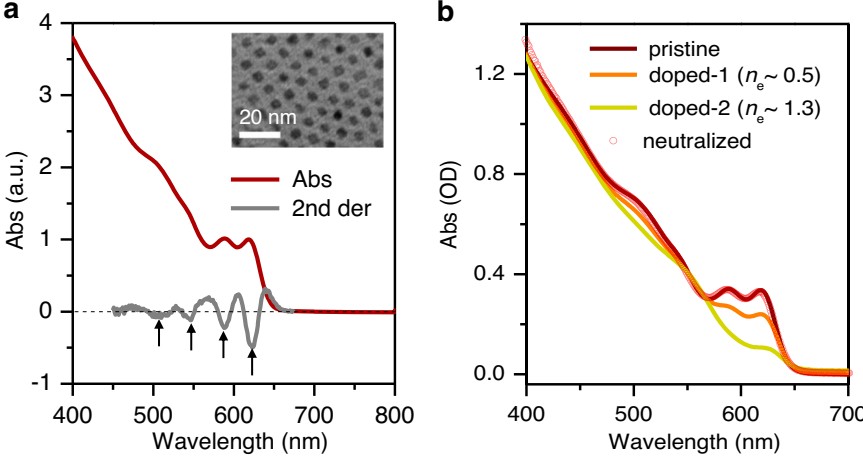

**Fig. 2 Optical properties of QDs. a** Absorption spectrum of 5.5-nm CdSe QDs (red) and its second derivative spectrum (gray). Four lowest energy transitions are indicated. Inset is a typical transmission electron microscope image of the QDs. **b** Absorption spectra of the pristine sample (wine), *n*-doped-1 (orange), *n*-doped-2 (yellow) and re-neutralized sample (red circles). The average numbers of doped conduction band electrons are ~0.5 and 1.3, respectively, for *n*-doped-1and *n*-doped-2.

electrons there are many electrons doped into the intragap trap states[45]. The important consequences of these trap-state electrons will be discussed later in our spectroscopy experiments. Photoluminescence of the pristine QDs was strongly quenched in the doped samples (Supplementary Fig. S3).

The *n*-doped samples can be neutralized upon exposure to the air and the absorption bleaching can be fully recovered (Fig. 2b). For this reason, all the *n*-doped samples used in this study were carefully deaerated. The neutral QDs used for comparison were obtained by exposing the doped samples to the air. This helps to exclude the impact of side effects (e.g., ligand desorption) arising from photochemical doping on the carrier dynamics of doped QDs.

**Hot electron dynamics in neutral QDs**. We applied pump-probe transient absorption (TA) spectroscopy to interrogate hot electron dynamics in QDs. The pump pulse was tuned in resonance with $1S_e$-$1S_{3/2,h}$ or $1P_e$-$1P_{3/2,h}$ transitions depending upon the experimental needs and the probe pulse was a broad-band white light continuum; details for the set-up has been described elsewhere[37]. The pump fluence was kept low to assure experiments were performed under single-exciton conditions (average exciton number per QD ~0.017; see Supplementary Fig. S4) and the samples were vigorously stirred during measurements.

We first investigated hot electron dynamics in neutral QDs as a benchmark. Figure 3a, b shows the TA spectra of 5.5-nm CdSe QDs pumped at the $1S_e$-$1S_{3/2,h}$ and $1P_e$-$1P_{3/2,h}$ excitons using 624 and 547 nm pulses, respectively. In the former case, the TA spectral formation is limited by the instrument response only and the spectral features show no evolution with time. Upon formation, these features are long-lived, showing 63% decay within a time window of 7.7 ns (Supplementary Fig. S5). The bleach features of $1S_e$-$1S_{3/2,h}$ (~618 nm) and $1S_e$-$2S_{3/2,h}$ (~589 nm) excitons arise from state-filling effect of the $1S_e$ electron. The derivative-like features from 480 to 570 nm contain not only state-filling of $1S_e$-$3S_{1/2,h}$ transition at 505 nm due to the $1S_e$ electron but also Stark effect features of $1P_e$-$1P_{3/2,h}$ arising from its Coulomb interaction with the pump-generated $1S_e$-$1S_{3/2,h}$ exciton[46,47].

In the case of $1P_e$-$1P_{3/2,h}$ excitation, we clearly observe spectral evolution near ~540 nm displaying decay of the $1P_e$-$1P_{3/2,h}$ bleach within 1 ps (Fig. 3b), in contrast to the observation of an isosbestic point at this wavelength in Fig. 3a. In addition,

there is a photoinduced absorption (PIA) feature at ~650 nm that can be ascribed to Coulomb interaction between the $1S_e$-$1S_{3/2,h}$ exciton and the pump-generated $1P_e$-$1P_{3/2,h}$ exciton inducing a redshift to the former[48]. This feature also decays within 1 ps as the exciton relaxes from $1P_e$-$1P_{3/2,h}$ to $1S_e$-$1S_{3/2,h}$. More specifically, it is the $1P_e$ to $1S_e$ relaxation because the TA spectrum of II–VI group core-only QDs is well known to be dominated by the electron[46,47].

Both 1P bleach and PIA features can be used to follow the relaxation dynamics of the $1P_e$ hot electron. As plotted in Fig. 3c, the kinetic traces probed at 542 nm (1P bleach) and 652 nm (PIA) follow the same decaying behavior, and both are complementary to the formation kinetics of the 1S bleach (618 nm). Simultaneous fitting of these kinetics reveals a relaxation time constant of 210 ± 40 fs. This ultrafast relaxation is consistent with previous reports[28]. Considering that the gap between $1P_e$ and $1S_e$ levels is ~270 meV, much larger than the longitudinal optical phonon energy of 25.5 meV in zinc blende CdSe QDs[49], a phonon bottleneck is expected. However, electron-to-hole energy transfer is strongly enhanced in QDs, leading to sub-ps hot electron relaxation.

**Spin blockade observed in lightly-doped QDs**. We then proceeded to study the effect of preoccupying electrons on hot electron dynamics in the *n*-doped-1 sample of 5.5-nm QDs. Figure 4a shows its TA spectra at indicated delays following $1S_e$-$1S_{3/2,h}$ excitation. The TA features are similar to those of neutral QDs. However, when we plot the 1S bleach kinetics and compare it with that of neutral QDs (Fig. 4b), we find that the bleach shows much faster recovery in the doped sample. This is because the preexisting electrons open up a nonradiative multi-carrier Auger recombination channel for the photoexcited exciton[50–52]. Because the doped sample also contains undoped QDs, we can normalize the 1S bleach kinetics of neutral and doped samples to a long-lived tail and then subtract the former from the latter to isolate the kinetics of pure doped QDs. Fitting the kinetics reveals an Auger-dominated lifetime constant of 460 ± 30 ps (Fig. 4b inset).

In principle, because *n*-doped-1 has a low doping level of 0.5 $1S_e$ electron per QD, the measured Auger lifetime should be that of a negative trion (X−). However, the lifetime of X− previously reported for 5.5-nm CdSe QDs coated with a thin ZnS shell is 740 ps[50]. The inconsistency lies in the ZnS shell that effectively

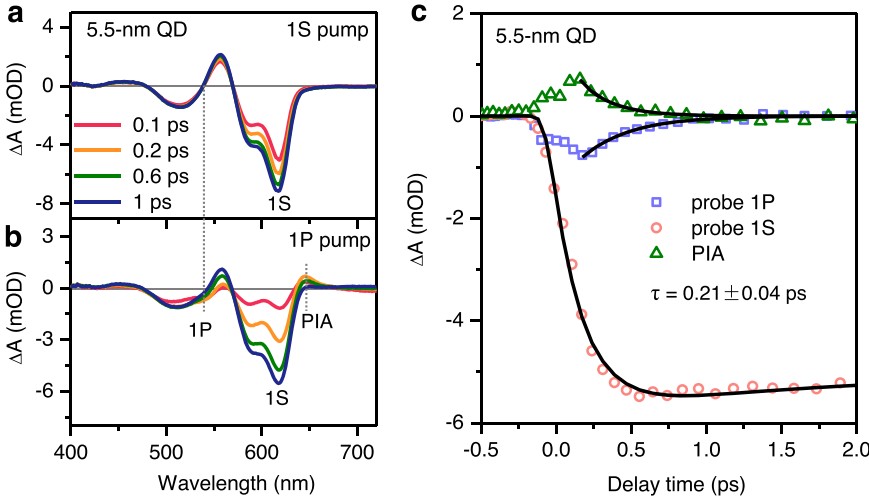

**Fig. 3 Hot electron dynamics in neutral QDs.** Transient absorption (TA) spectra of neutral CdSe QDs probed at indicated time delays following excitations at **a** $1S_e$-$1S_{3/2,h}$ and **b** $1P_e$-$1P_{3/2,h}$ transitions by 624 and 547 nm pulses, respectively. The 1S bleach, 1P bleach and PIA features are indicated in **b**. **c** TA kinetics probed at the 1S bleach (~617 nm; red circle), 1P bleach (~544 nm; blue square) and PIA (~646 nm; green triangle) features in the case of $1P_e$-$1P_{3/2,h}$ excitation. Black solid lines are simultaneous fits to a single-exponential relaxation time constant of 0.21 ps.

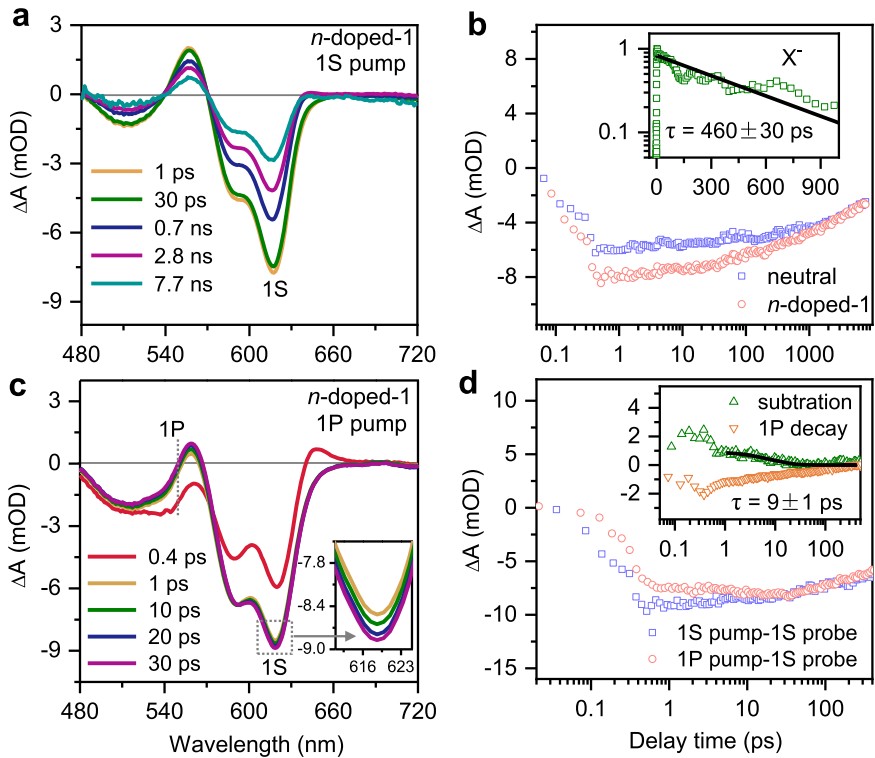

**Fig. 4 Spin blockade in lightly *n*-doped QDs.** TA spectra of *n*-doped-1 probed at indicated time delays following excitation at **a** $1S_e$-$1S_{3/2,h}$ and **c** $1P_e$-$1P_{3/2,h}$ transitions by 624 and 547 nm pulses, respectively. Inset in **c** is an enlarged view of the slowly growing 1S bleach feature. **b** TA kinetics probed at the 1S bleach for *n*-doped-1 (red circle) and neutral (blue square) samples, scaled at their slowly-decaying tail, in the case of $1S_e$-$1S_{3/2,h}$ excitation. Inset is the Auger recombination kinetics (green square) obtained by performing a subtraction between the two traces in the main panel and its single-exponential fit to a time constant of 460 ps. **d** TA kinetics probed at the 1S bleach for *n*-doped-1 under $1S_e$-$1S_{3/2,h}$ (blue square) and $1P_e$-$1P_{3/2,h}$ (red circle) excitations, scaled at their slowly-decaying tail. Inset are the 1S kinetics obtained by performing a subtraction between the two traces in the main panel (green triangle) and the kinetics monitored at the 1P bleach (orange triangle). Black solid line is a fit to the slow relaxation component with a time constant of 9 ps indicative of a spin blockade.

alleviates trap states on CdSe surfaces. In our core-only QDs, surface states, especially electron-trapping states, can be filled by electrons during photochemical doping[45]. These trap-state electrons could accelerate the rate of Auger recombination as this rate scales up with the number of carriers[53].

Figure 4c shows the TA spectra of *n*-doped-1 following $1P_e$-$1P_{3/2,h}$ excitation. In this case, we observe spectral evolution near 544 nm related to decay of the 1P bleach feature. In Fig. 4d, we compare the 1S bleach kinetics of *n*-doped-1 under 1S and 1P excitations. They share the same decaying kinetics, with the

decay reflecting Auger recombination of charged single-excitons in the doped QDs and recombination of neutral single-excitons in the undoped QDs in the $n$-doped-1 ensemble sample, but in the case of 1P excitation there is a slow growth process persisting until ~30 ps. By taking the difference between the two traces, we obtain 1S bleach formation kinetics in the doped sample which is consistent with the kinetics monitored at the 1P bleach (Fig. 4d inset). The slow relaxation component can be fitted to a time constant of $9 \pm 1$ ps. The fast relaxation component is contributed by undoped QDs in the ensemble as well some doped QDs in which the preexisting $1S_e$ electron and the photoexcited $1P_e$ electron have the opposite spin.

Specifically, the average number of doped electrons per QD is ~0.5 for the $n$-doped-1 sample. Assuming a Poisson distribution of the doped electrons[38], ~60% of the QDs were undoped, ~30% were doped with one electron, and ~10% were doped with ≥2 electrons. QDs doped with ≥2 electrons would not display 1S exciton bleach on the TA spectra because the absorption was already fully blocked. Among the QDs doped with one electron, statistically only half of the photoinjected $1P_e$ electrons would have the same spin as the doped $1S_e$ electron due to random distribution of the spin directions. Note that even if we create spin-polarized $1P_e$ electrons using circularly-polarized pump pulses, this statement still holds because there is no control over the spin directions of pre-doped $1S_e$ electrons. Thus, the portion of the 1S bleach amplitude that would correspond to the spin blockade effect is only 1/6 ($=1/3 \times 1/2$) of the total 1S bleach amplitude. According to Fig. 4d, the total 1S bleach amplitude under 1P pump is ~−8 mOD, and therefore the spin blockade signal can be estimated as ~−1.3 mOD, which is consistent with the amplitude of the slow formation component (~−1.2 mOD) observed in Fig. 4d.

The agreement between the estimated and measured signal amplitudes above provides strong evidence that the slow $1P_e$ relaxation component of 9 ps can be ascribed to a Pauli spin blockade induced by the preexisting $1S_e$ electron (Fig. 1b left). In QDs where the photoinjected $1P_e$ electron and the pre-doped $1S_e$ have the same spin direction, either the $1P_e$ or $1S_e$ electron has to flip its spin before the $1P_e$ electron can relax down. For CdSe QDs, we believe it should be the $1P_e$ spin flip for the following reasons. The Bloch functions of conduction band edge levels of CdSe are mainly consisted of 5s atomic orbitals of Cadmium showing weak spin-orbital coupling. On the other hand, the envelope wavefunction of a $1P_e$ electron has an angular momentum of 1 that can be transferred to the spin whereas a $1S_e$ electron does not. Interestingly, a recent study of pump-power dependent TA kinetics of CdSe/CdS core/shell QDs also implies a spin blockade for $1P_e$ electron relaxation with a time constant of 25 ps;[54] this is longer than the time constant in our sample likely because the shell can slow down spin flip. It is worth mentioning that, in that study the spin-blockade was realized by transiently injecting the QDs with excitons, whereas herein we observe the effect in free-standing $n$-doped QDs that are stable in the steady state.

Although the population analysis above as well as comparison to literature strongly supports the assignment of the 9 ps slow relaxation to a Pauli spin blockade, an alternative possibility is that the nonradiative recombination between the photoexcited hole and pre-doped trap-state electron occurred on a sub-ps timescale, thus eliminating the hole before the electron-to-hole energy transfer. This would lead to a situation similar to the one depicted in the rightmost panel of Fig. 1b, i.e., a long-lived $1P_e$ electron enabled by a phonon bottleneck. However, as we will present later, this type of $1P_e$ electron will have a lifetime of 100 s of ps.

**Phonon bottleneck observed in heavily-doped QDs.** Encouraged by the results in lightly-doped QDs, we continued to measure hot electron dynamics in heavily-doped 5.5-nm QDs ($n$-doped-2) to test the prediction in Fig. 1b (middle), that is, very long-lived $1P_e$ electron when the $1S_e$ level is fully occupied by two electrons. Figure 5a shows its TA spectra at indicated delays following $1S_e$-$1S_{3/2,h}$ excitation. Note that, however, this excitation only excites QDs doped with one electron because QDs doped with two or more electrons cannot absorb the 1S resonant pump photon. Surprisingly, the pump-induced TA features rapidly recover on the ps timescale. The decay is not single-exponential and can be fitted to a two-exponential decay function with a major component (78%) of 1 ps and a minor one (22%) of 31 ps; the amplitude-averaged lifetime is 7.6 ps (Fig. 5b). This ultrafast decay is also a consequence of Auger recombination induced by doped electrons. Although the nominal number of doped $1S_e$ electron is 1.3 in $n$-doped-2, there should be many more electrons doped into the trap states that strongly accelerate Auger recombination (Fig. 5b inset). It is not clear yet why the Auger recombination in $n$-doped-2 is so much faster than that in $n$-doped-1. In principle, all the trap states have to be filled before the electrons can be doped into the conduction band edge, and hence, the numbers of trap-state electrons should be similar in both samples. Further, the numbers of band edge electrons in both samples are not strongly different either (0.5 versus 1.3), contradicting the orders-of-magnitude difference in their Auger lifetimes. One possibility is that the densities of trap states are not constant but rather vary with the doping conditions. The strong reducing reagent, $LiEt_3BH$, could perturb the binding between QDs and surface ligands, thus introducing additional trap states to the QDs. Because the amount of $LiEt_3BH$ applied to $n$-doped-2 is much larger than $n$-doped-1, we suspect that there are many more surface-state electrons in the former than the latter.

Interestingly, when we pump the $n$-doped-2 sample at the $1P_e$-$1P_{3/2,h}$ transition, we observed peculiar spectral features associated with hot electron dynamics. As shown in Fig. 5c, the TA spectra display an absorptive rather than bleaching feature in the transition range of $1S_e$-$1S_{3/2,h}$ and $1S_e$-$2S_{3/2,h}$ excitons. This means photoexcitation leads to enhanced absorption from these excitons. The build-up time of this signal (Fig. 5d) is consistent with the time constant of the major component of the Auger recombination process measured in Fig. 5b. The minor, slow component of Auger recombination is not reflected on the build-up kinetics in Fig. 5d because of a convolution between this slow formation and some fast decay components that will be explained later. As schematically depicted in Fig. 5d inset, Auger recombination annihilates a $1S_e$ electron, leading to temporary enhancement of the 1S exciton absorption. The enhanced 1S absorption eventually decays because a $1P_e$ electron will relax to the $1S_e$ level. Notably, however, the enhanced absorption is remarkably long-lived, persisting into nanosecond timescale. Consistently, the bleach feature at 555 nm that can be assigned to the $1P_e$ electron is also long-lived (Fig. 5c). Fitting the recovery kinetics of 1S absorption and 1P bleach indicates that the time constant for $1P_e$-to-$1S_e$ relaxation is $320 \pm 10$ ps (Fig. 5d). We ascribe the very slow hot electron relaxation to the phonon bottleneck effect. As shown in Fig. 5d inset, the Auger recombination annihilates not only a $1S_e$ electron but also a valence band hole, disabling the electron-to-hole energy transfer pathway for hot electron relaxation. Our previous work on copper-doped CdSe QDs demonstrates that hot electron lifetime can be significantly prolonged by trapping the hole to the copper dopants[37]. However, the weak yet still existing coupling between a hot electron and a copper-localized hole limits the hot electron lifetime to below 10 ps. The current system represents a

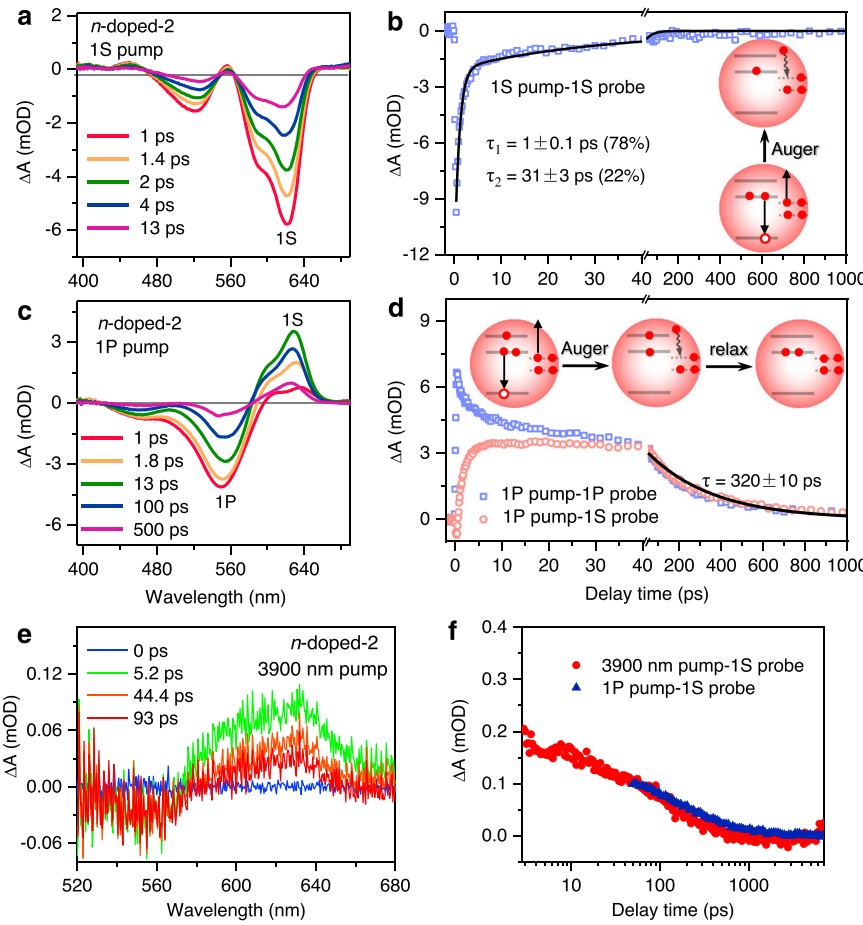

**Fig. 5 Phonon bottleneck in heavily *n*-doped QDs.** TA spectra of *n*-doped-2 probed at indicated time delays following excitations at **a** $1S_e$-$1S_{3/2,h}$ and **c** $1P_e$-$1P_{3/2,h}$ transitions by 624 and 547 nm pulses, respectively. The 1S absorption and 1P bleach features are indicated in **c**. **b** TA kinetics probed at the 1S bleach for *n*-doped-2 under $1S_e$-$1S_{3/2,h}$ excitation (blue square) and its bi-exponential fit to time constants of 1 and 31 ps (black line). The scheme on the right is a depiction of the trap-electron-assisted Auger recombination. Note that only QDs doped with one electron are shown because QDs doped with two or more electrons cannot absorb the 1S resonant pump photon. **d** TA kinetics probed at the 1S absorption (red circle) and 1P bleach (blue square; inversed and scaled) for *n*-doped-2 under $1P_e$-$1P_{3/2,h}$ (547 nm) excitation. Black solid lines are simultaneous fits of their slow relaxation components to a time constant of 320 ps indicative of a phonon bottleneck. Inset is a schematic depiction of trap-electron-assisted Auger recombination followed by slow hot electron relaxation in heavily *n*-doped QDs. Only QDs with two doped electrons are shown in this scheme. In QDs doped with one electron, there is a competition between Auger recombination and spin-flip hot electron relaxation; see main text for details. **e** TA spectra at indicated delays of an *n*-doped-2 sample under 3900 nm pump. **f** Comparison of TA kinetics probed at the 1S absorption feature under 3900 nm pump (red circles) and 1P exciton pump (blue triangles).

significant progress compared to copper-doped QDs because it completely switches off the electron-to-hole energy transfer pathway.

There are fast decay components for the 1P bleach (Fig. 5d), resulting from inhomogeneous distribution of the predoped electrons in the ensemble sample. The average number of electrons doped per QD in the heavily-doped sample is ~1.3. Using Poisson statistics, the percentages of QDs doped with 0, 1, and ≥2 electrons are 27.2%, 35.4%, and 37.3%, respectively. In QDs doped with ≥2 electrons, the photoinjected $1P_e$ electron cannot relax until a doped $1S_e$ electron is consumed by Auger recombination; thus all these QDs should display long-lived hot electrons. For QDs doped with one electron, half of them should display rapid $1P_e$ relaxation (no spin blockade), whereas the other half should experience the spin blockade. The spin-flip process (time constant ~9 ps) competes with the Auger recombination (amplitude-averaged time constant ~7.6 ps) and this competition determines the partition between QDs that relax via spin-flip and those showing the phonon bottleneck (45.8%:54.2%). In the undoped QDs, all the $1P_e$ hot electrons rapidly relax via electron-to-hole energy transfer. Thus, the overall percentage of QDs displaying long-lived hot electrons is

estimated as: $37.3\% + 35.4\% \times 54.2\%/2 = 46.9\%$. According to Fig. 5d, the percentage of the long-lived components of the 1P kinetics is ~44.8% (3.0 mOD/6.7 mOD), which is in good agreement with the estimated ratio.

It is interesting to examine the fate of the surface-doped electron that is excited in an Auger recombination event (Fig. 5b scheme). One possibility is that it is transiently ejected outside the QD after accepting the electron-hole recombination energy (i.e., Auger ionization) and eventually returns to the surface states. The other possibility is that the electron is excited to a very high level either in the conduction band or in the surface states. In this case, if the electron relaxes back to the band edge, we should expect slow growth again of the 1S bleach as a manifestation of the phonon bottleneck effect, which, however, is not observed in our experiment (Fig. 5b). Therefore, we suspect that the highly-excited electron also returns to the intra-gap surface states, but the "invisibility" of the electrons in the surface states prohibits conclusive statements on the relaxation pathways of the excited electron.

Although the surface-state electrons and associated ultrafast Auger recombination have enabled the observation of a phonon

bottleneck, they could also complicate the real application of $n$-doped QDs. However, the surface-state electrons are not necessarily required. As depicted in Fig. 1b (right), the phonon bottleneck can be directly realized using a mid-IR-pump, visible-probe TA experiment[55,56] (see "Methods"). In this experiment, we chose a pump wavelength of 3900 nm which is roughly in resonance with the $1S_e$ to $1P_e$ intraband transition. As shown in Fig. 5e, following this intraband pump, a positive 1S exciton absorption band was induced. This is because the predoped electrons that initially occupied the $1S_e$ level were partially promoted to the $1P_e$ level, increasing the interband 1S exciton absorption. The 1S absorption decays within a few 100 s of ps (Fig. 5f), in reasonable agreement with the hot electron relaxation dynamics probed by visible-pump, visible-probe TA.

It is important to note that previous studies uncovered a universal mechanism of nonadiabatic interaction with surface ligands that can break the phonon bottleneck for hot carrier relaxation in QDs[30,33,57]. In this mechanism, the carrier wavefunction that is delocalized to QD surfaces interacts with the nuclear degree of freedom of surface ligands, inducing nonadiabatic relaxation within quantum-confined states. This mechanism is reported to be responsible for sub-ps hot hole relaxation in CdSe QDs[30,57] and hot electron and hole relaxation in strongly-confined $CsPbBr_3$ perovskite QDs[33]. The fact that we can observe 320-ps-long hot electron here implies that this mechanism is also suppressed in heavily $n$-doped CdSe QDs. This is likely related to the large number of electrons doped into surface trap states. Repulsive interaction between these surface electrons and the core electron confines the latter to the core and hence suppresses nonadiabatic interaction between the core electron and surface ligands.

**Size dependence and comparison to other materials**. We measured hot electron dynamics in lightly and heavily doped QDs of other sizes and found that the hot electron relaxation dynamics were largely independent on QD size; see Supplementary Figs. S6–S12. The spin blockade results in a hot electron lifetime of ~10 ps for all the lightly-doped samples (Fig. 6a), whereas the phonon bottleneck gives rise to a hot electron lifetime of ~300 ps for all the heavily-doped samples (Fig. 6b). The spin blockade effect is independent on QD size because the spin-flip time of a $1P_e$ electron should be dictated by the strength of spin-orbital coupling which is not obviously correlated with QD size. The phonon bottleneck effect, in principle, should display a size-dependence because the gap between $1P_e$ and $1S_e$ levels becomes larger in smaller QDs, which in turn requires more phonons to participate in hot electron relaxation if it occurs via the weak multi-phonon mechanism. However, because of the limited range of QD sizes investigated here, this effect is not strong enough to significantly alter the hot electron lifetime. In our experiments, heavy doping of small-size (<4.0 nm) CdSe QDs was unsuccessful due to stability issues.

Last but not the least, we compare the hot electron lifetime achieved in this work to those of other systems reported in the literature. These systems include various lead halide perovskite materials whose hot carrier dynamics are intensively studied in recent years[11,12] and our recently reported copper-doped CdSe QDs[37]. Because hot carrier lifetime depends on the excessive energy of hot carriers, we adopt a more quantitative descriptor, the hot carrier energy loss rate ($dE/dt$). It can be calculated from the excessive energy of hot carriers above their band edges used in specific experiments and the reported hot carrier lifetime[37]. As plotted in Fig. 6c, $dE/dt$ of lightly $n$-doped QDs (0.03 eV/ps) is 2.5-fold slower than that of copped-doped QDs (0.075 eV/ps), and both are much slower than those of all the perovskite

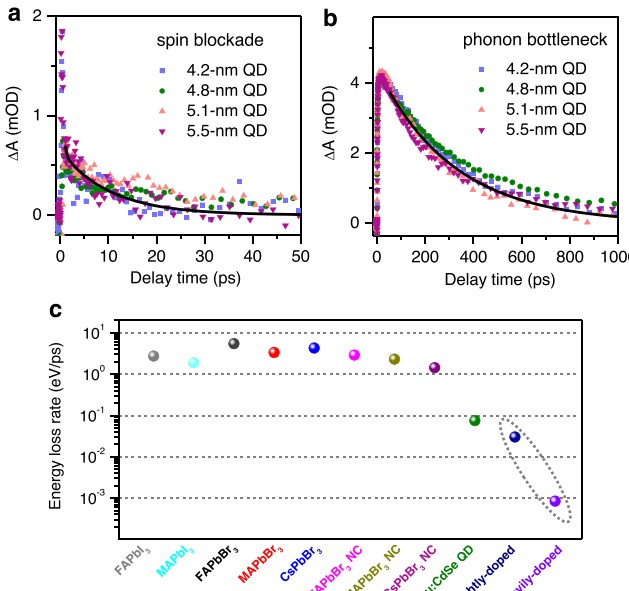

**Fig. 6 Size dependence and comparison to other materials.** Hot Electron relaxation in **a** lightly and **b** heavily $n$-doped QDs of varying sizes. **c** Hot carrier energy loss rates calculated for $FAPbI_3$, $MAPbI_3$, $FAPbBr_3$, $MAPbBr_3$, and $CsPbBr_3$ bulk films, $FAPbBr_3$, $MAPbBr_3$, and $CsPbBr_3$ NCs, copper-doped CdSe QDs and lightly and heavily $n$-doped CdSe QDs. The last two samples (circled) are from this study, whereas the rest are adapted from the literature[37].

materials (≥1.44 eV/ps). More importantly, heavily $n$-doped QDs display a slow hot electron cooling rate of 0.84 meV/ps due to a phonon bottleneck effect, representing 88-fold improvement compared to previous copper-doped QDs and three orders of magnitude improvement than undoped QDs or perovskite materials.

## Discussion

Importantly, the phenomena of spin blockade and phonon bottleneck can also observed in $n$-doped solid-state films of CdSe QDs (Supplementary Fig. S13), suggesting the possibility of utilizing the hot electrons in devices. In particular, QD-sensitized metal oxide (such as $TiO_2$) films offer a promising platform for hot electron extraction from QDs owing to the strong electronic coupling between QDs and metal oxides as well as the large density of electron-accepting states in the conduction band of metal oxides. For example, electron transfer from CdSe QDs to mesoporous $SnO_2$ films was reported to occur in a few picoseconds;[58] electron transfer from PbS and PbSe QDs to $TiO_2$ could be promoted to the femtosecond timescale by engineering the interfacial coupling[24,59]. Our prior study also showed evidence for hot electron transfer from copper-doped CdSe QDs (hot electron lifetime ~8.6 ps) to $TiO_2$. Such hot electron extraction should be efficient with the heavily $n$-doped QDs here featuring a hot electron lifetime as long as 300 ps.

In conclusion, we have studied hot electron dynamics in $n$-doped CdSe QDs. This unique system allows us to exploit carrier-carrier, carrier-phonon and spin-spin interactions in QDs to observe very long-lived hot electrons. For lightly $n$-doped QDs we observe a $1P_e$ hot electron lifetime of ~10 ps as a manifestation of a Pauli spin blockade arising from preoccupying $1S_e$ electrons. For heavily $n$-doped QDs, a large number of electrons residing in the surface states induce ps Auger recombination that annihilates the valance band hole. This allows us to observe 300-ps-long $1P_e$ hot electrons as a manifestation of a phonon bottleneck effect,

representing three orders of magnitude lengthening as compared to typical QDs or bulk semiconductors. These results suggest numerous opportunities of hot electron harvesting from *n*-doped colloidal QDs.

## Methods

**Synthesis of CdSe QDs**. CdSe QDs were synthesized using a previously reported "heat-up" method[60]. In a typical synthesis, a mixture of 0.135 g of cadmium oleate, 0.022 g of selenium dioxide, and 8.8 g of ODE was loaded into a three-neck flask and was degassed under vacuum at 55 °C for 45 min. The mixture was then heated at 235 °C under $N_2$ atmosphere and aliquots were taken from the mixture to monitor the growth of QDs. When the lowest energy peak of QDs reached 591, 604, 611, or 618 nm, the reaction was stopped and QDs were precipitated from the mixture by adding ethanol. The precipitants were re-dispersed in toluene and precipitated by ethanol. This was repeated for three times and the final products were dispersed in toluene for optical measurements.

**Photochemical doping of CdSe QDs**. Photochemical doping experiments followed reported procedures[38,45,53]. Briefly, $LiEt_3BH$ (1 M solution in tetrahydrofuran) was diluted to 0.01 M with toluene. The CdSe QDs solution with an optical density of ~0.34 at the band-edge peak was transferred into a glove box with a $N_2$ atmosphere (oxygen level <0.1 m). The diluted $LiEt_3BH$ solution was added into the QDs-toluene solution under vigorous stirring and under room light. The solution was then illuminated by a UV lamp to accelerate the reaction until it reached equilibrium. For successful photochemical doping, 10-to-100 equivalents of $LiEt_3BH$ per QD were required. The degree of doping could be controlled by varying the amount of $LiEt_3BH$ used in the reaction. The doped QDs solution was sealed in a custom-made airtight cuvette (optical path 1 mm) and transferred out of the glove box for all optical measurements. The neutralized solution was obtained by exposing the doped QDs solution to the air for a few minutes. The QD films for doping were prepared by drop-casting QD solution onto glass substrates. The films were then treated with 0.01 M $LiEt_3BH$-toluene solution and also illuminated by a UV lamp in the glove box with a $N_2$ atmosphere (oxygen level <0.1 ppm). Finally, the doped QD films were encapsulated under $N_2$ condition using custom-made airtight containers and then transferred out for optical measurements.

**Transient absorption**. Femtosecond visible pump-probe TA measurements were based on a regenerative amplified Ti:sapphire laser system (Coherent; 800 nm, 70 fs, 6 mJ/pulse, and 1 kHz repetition rate) as the laser source; details were described elsewhere[37]. As for the mid-IR pump-visible probe experiment, the pump light at 3900 nm was generated via difference-frequency-generation using a Pharos laser (1030 nm, 100 kHz, 230 fs pulse-duration; Light Conversion) and an OPA. The other set-ups are similar to the visible pump-probe TA.

## Data availability

The experiment data that support the findings of this study are available from the corresponding author upon reasonable request.

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

## Acknowledgements

We acknowledge financial support from the Ministry of Science and Technology of China (2018YFA0208703), the Strategic Pilot Science and Technology Project of Chinese Academy of Sciences (XDB17010100), the National Natural Science Foundation of China (21773239, 21973091, 51961165109) and the LiaoNing Revitalization Talents Program (XLYC1807154).

## Author contributions

K.W. conceived the ideas and designed the project. L.W. and T.D. synthesized the samples. J.W. characterized the samples, performed spectroscopy experiments and analyzed the data. S.Y. helped with the mid-IR experiments. K.W., J.W., and D.X. wrote the paper with contributions from all authors.

## Competing interests

The authors declare no competing interests.
