## [Peer Review File · Nature Communications]

REVIEWER COMMENTS

Reviewer #1 (Remarks to the Author):

Wu et al, have demonstrated the observation of slow hot electron relaxation time in n-doped colloidal quantum dots (QDs) thanks to spin blockade and phonon bottleneck. They found a slow hot electron relaxation of ~ 10 ps in lightly doped QDs due to Pauli spin blockade and ~ 300 ps for hot electron relaxation in heavily doped QDs due to phonon bottleneck in the presence of Auger-assisted hole scavenger. Such observation is interesting and would be useful for photon-electron conversion applications. However, several issues need to be further addressed.

1. To prolong the hot electron relaxation by phonon bottleneck, the removal of photo-excited holes is a prerequisite process, here by Auger recombination with the electrons residing in the surface states. However, for practical utilization of hot electrons in solar energy related applications, those surface states would play negative roles, for example, blocking charging injection and enhancing charge trapping. In addition, the photo-excited electrons cannot be stably occupied in the surface states, please comment on those issues.
2. Please compare steady-state PL spectra for QDs with and without photochemical doping.
3. Is it possible to observe those interesting behavior, namely slow hot electron relaxation in the solid state?
4. How to evidence the population of surface states after photochemical doping?
5. Please also make some comments on the charge extraction of hot electrons from colloidal QDs to metal oxides, which is critical for practical utilization in photon-electron conversion.

Reviewer #2 (Remarks to the Author):

In this study, Wang et.al. investigate ultrafast hot carrier dynamics in photochemically-doped CdSe quantum dots (QDs). They claim a spin-block effect for relatively-low doped QDs (~ 0.5 electrons in the 1Se state), which prolonged the 1Pe to 1Se hot carrier decay time from sub-ps to nearly 10 ps. For highly-doped QDs, they observe that some fraction of hot carriers can undergo very slow cooling with a decay lifetime exceeding ~ 300 ps. They explain their data by a phonon bottleneck effect for hot electrons following Auger recombination, which removes the photogenerated hole states. The paper deals with an important topic which could trigger interests from a broad range of audience. Data for highly doped QDs are fascinating (in figure 5 c-d) and new. However, I do not recommend the paper for Nature communications, because of (1) no clear evidence for spin-blockade effect for low-doped QDs based on their results, and (2) insufficient data analysis and discussion to support the photon bottleneck effect. Some of my major comments are the following:

(1) For the low-doped QDs, the only experimental observation that seems to be consistent with spin-blockade is the relatively prolonged hot carrier relaxation time. However, other effects could also lead to slow cooling by electron doping. For instance, if the photo-excited holes recombine with the electrons injected into defect states non-radiatively, the hot electrons can also be long-lived as the electron-hole energy transfer pathway is suppressed. The authors have claimed a critical role of defects for the highly-doped QDs, but completely neglect this for the low-doped ones.

Therefore, for a clear experimental demonstration of spin-blockade effect, some further discussions or even experimental pieces of evidence are required: (a) as the doped electrons are not spin-polarized, and photogenerated electron and hole are produced by a linearly-polarized pulse, 50% of doped QDs should exhibit the spin-blockade effect (if it is active). Can the author do further data analysis to check this point? In principle, this can be done by applying e.g. Poisson statistics for photoexcitation. The normalized dynamics shown in fig. 4 (b) and (d) do not provide sufficient information on the fraction of charge carriers following the slow decay channel. In fact, by closely looking at the dynamics,

the slow decay claimed by the authors is only a small portion of the dynamics. The majority of charge carriers, even following 1P excitation, fill the 1Se states in sub-ps. (b) to fundamentally prove spin-blockade, further experiments based on spin-polarized electrons injected into 1Se states are encouraged. By further changing the circular polarization of the pump, one should see a drastic change in hot carrier lifetimes. While the comment (b) seems challenging, further discussions for (a) is mandatory.

Finally, the spin-blockade concept is not new and has been proposed and demonstrated previously (J. Phys. Chem. Lett. 2019, 10, 2341–2348);

(2) For highly-doped QDs, while a slow decay in the dynamics over 300 ps seems clear from the dynamics, a critical point that is relevant for application is what percentage of charge carriers follow this path? Some quantitative analysis is necessary (e.g. by applying Poisson statistics for the carrier population following excitation) based on the signal strength (not just the normalized dynamics).

(2a) There is a large portion of fast decay for the 1P bleach, which the author attributed to "both Auger recombination and spin-flip relaxation". Why is that? Why did these effects not matter for the low-doped QDs?

Furthermore, I found some claims for discussion are not well supported or inconsistent for the main conclusion of "phonon bottleneck" effect:

(2b) For the scheme in the figure 5b, what happened to the electrons in the defect following the Auger process following 1S excitation? In the scheme, this electron just disappeared and the authors did not comment on this. The electron can be excited to high energy states following Auger excitation, and should stay hot following the "phonon bottleneck" scenario proposed in the paper. However, the dynamics show only an ultrafast decay.

For the scheme in the figure 5d, before photoexcitation there is 1 electron in 1S state, and arbitrarily 4 electrons in the defects. In the end, following Auger and relaxation, the system did not decay back to the initial states with only 3 electrons in the defects. Why is that?

In any case, following what has been shown in both schemes in figure 5 b and d, one should expect long-lived hot carriers in both cases. But this is not in line with experimental results.

(2c) Auger rates throughout the paper: for the low-doped QDs, the authors reported an Auger lifetime of 460 ps. This value is way higher than what has been reported previously (in the range of tens ps for the same diameter, see in figure 3 in the paper by Klimov et. al. Science 2000, 1011-1013). Why is that? Can the dynamics be interpreted differently? Can we normalize the dynamics to the early time (rather than the later time scale)? By doing so, we will see a faster decay for doped QDs in the range of 100-1000 ps, which may be attributed to e.g. trapping.

In contrast, for the highly-doped QDs, the Auger lifetime is as short as 1 ps. How and why the Auger lifetime can be tuned more than two orders of magnitude by adding ~ 0.8 more electrons/QDs (by comparing low (0.5 1Se electron) and high doping (1.3 1Se electron)? Based on the paper from Klimov (Science 2000, 1011-1013), compared to the first order Auger process, the rates for high order Auger processes should increase, but not as dramatic as reported here. Although the authors claimed that "there should be many more electrons doped into the trap states that strongly accelerate Auger recombination", this should also apply to the low-doped case. The defects must be filled before they can host electrons in the 1Se states (0.5 for low-doped case).

This is a critical point, as a few discussions made by the authors are based on such assignments in dynamics: e.g. "Note that there is a fast decay component for the 1P bleach that has complicated origins from both Auger recombination and spin-flip relaxation."

(2d) In the introduction, the authors claimed that "Alternatively, a doped 1Se electron can be directly excited to the 1Pe level using a mid-infrared photon; the resulting 1Pe electron is fully decoupled from any electron or hole and it should be very long-lived because of a phonon bottleneck (Fig. 1b, right)". While it is a bit misleading (as the authors did not follow such an intra-band excitation method later in the paper), I find this statement fascinating: it provides a clear route for demonstrating the phonon bottleneck effect. However, in reality, the author follows an "inter-band" excitation scheme, which

complicates the data analysis. To fully support the conclusion of the phonon bottleneck effect, I would recommend the author to conduct such studies. It should not be that difficult, as one needs just to excite the doped QDs below their optical bandgap.

Other comments to the paper:

(3) For low-doped QDs, in the data analysis part, subtraction of two dynamics following 1S and 1P excitations have been applied. This implies a common decay pathway shared by 1P and 1S excitation. What is the origin of the common dynamics?

(4) On page 9, the authors stated that "This decay annihilates a 1Se electron, deviating from our initial expectation that the 1Pe electron would be long-lived due to full occupation of the 1Se level." This statement is very confusing. Here you have excited the system only to the 1Se states. Why will this lead to long-lived 1Pe (because you did not excite it)?

(5) In the introduction, the authors claimed that "Hot electrons are unthermalized electrons that carry large kinetic energies." This is not correct, as hot electrons can be thermalized or non-thermalized. In fact, the connotation 'hot' implies there is a temperature associated with the electron distribution. Some rephrasing is needed.

(6) On page 10, the authors mentioned that "The build-up of this signal is on a few ps timescale (Fig. 5d), which is similar to the Auger recombination time shown in Fig. 5b."

What is exactly the build-up time? Why is it faster than the Auger time (~ 1 ps) for 1S pump? Given that the same Auger processes occur following both 1S and 1P excitations sketched by the authors, I would expect the Auger time is the same here for both 1S and 1P excitations.

Reviewer comments in black, our responses in red and revisions in blue

REVIEWER COMMENTS

Reviewer #1 (Remarks to the Author):

Wu et al, have demonstrated the observation of slow hot electron relaxation time in n-doped colloidal quantum dots (QDs) thanks to spin blockade and phonon bottleneck. They found a slow hot electron relaxation of ~ 10 ps in lightly doped QDs due to Pauli spin blockade and ~ 300 ps for hot electron relaxation in heavily doped QDs due to phonon bottleneck in the presence of Auger-assisted hole scavenger. Such observation is interesting and would be useful for photon-electron conversion applications. However, several issues need to be further addressed.

Response: We thank the reviewer very much for his/her kind comments on our work. Below we provide our point-to-point responses and revisions to address his/her remaining concerns.

1. To prolong the hot electron relaxation by phonon bottleneck, the removal of photo-excited holes is a prerequisite process, here by Auger recombination with the electrons residing in the surface states. However, for practical utilization of hot electrons in solar energy related applications, those surface states would play negative roles, for example, blocking charging injection and enhancing charge trapping. In addition, the photo-excited electrons cannot be stably occupied in the surface states, please comment on those issues.

Response: We thank the reviewer for these insightful comments. We also agree with the reviewer that Auger recombination and surface states are in general harmful to the real applications of QDs. In the photoexcited n-doped CdSe QDs, ultrafast Auger recombination induced by surface states removed the photogenerated hole and hence shut down the electron-to-hole energy transfer pathway. This accidental finding enabled us to observe the phonon bottleneck for hot electron relaxation QDs.

Nonetheless, as we discussed in the introduction on the rightmost panel of Fig. 1b, the phonon bottleneck in n-doped CdSe QDs should also be observable by directly exciting the doped electron from the 1Se to the 1Pe level using a mid-IR photon, without involving a valence band hole. This was not done in our prior study due to the difficulties in mid-IR transient absorption (TA) experiments compared to visible TA. We have now managed to perform a mid-IR-pump, visible-probe experiment for an n-doped sample (details provided in the revised Methods).

As shown in Fig. R1a, following 3900 nm pump which is roughly in resonance with the 1Se to 1Pe intraband transition, a positive 1S exciton absorption band was induced. This is because part of the doped electrons that initially occupied the 1Se level were promoted to the 1Pe level, thus increasing the interband 1S exciton absorption. The 1S absorption decays within a few 100s of ps (Fig. R1b), in reasonable agreement with the hot electron relaxation dynamics probed by

visible-pump, visible-probe TA.

Fig. R1. (a) TA spectra at indicated delays of an n-doped sample under 3900 nm pump. A 1S exciton absorption feature is induced. (b) TA kinetics probed at the 1S absorption feature under 3900 nm pump (red circles) and 1P exciton (547 nm) pump (blue triangles).

Revision: At the end of Page 14, we add the following contents to describe the mid-IR pump experiment:

“The surface-state electrons and associated ultrafast Auger recombination are not necessarily required for the observation of phonon bottleneck. The scheme in Fig. 1b (right) can be directly realized using a mid-IR-pump, visible-probe TA experiment^{54,55} (see Methods). As shown in Supplementary Fig. 6, following 3900 nm pump which is roughly in resonance with the 1Se to 1Pe intraband transition, a positive 1S exciton absorption band was induced. This is because the predoped electrons that initially occupied the 1Se level were partially promoted to the 1Pe level, increasing the interband 1S exciton absorption. The 1S absorption decays within a few 100s of ps (Supplementary Fig. 6), in reasonable agreement with the hot electron relaxation dynamics probed by visible-pump, visible-probe TA.”

2. Please compare steady-state PL spectra for QDs with and without photochemical doping.

Response: We thank the reviewer for this suggestion. The PL spectra of two n-doped samples with similar doping levels as prior n-doped-1 and n-doped-2 have been measured (Fig. R2). The PL was strongly but not completely quenched by the doped electrons. In principle, however, the PL of doped QDs should be completely quenched due to the fast Auger recombination. We therefore attribute the residual emissions to sub-sets of undoped QDs in the doped samples.

Fig. R2. Photoluminescence spectra of two n-doped samples with nominal average numbers of band-edge electrons of 0.45 (black) and 1.2 (red), in comparison to that of the neutralized sample (blue).

Revision: On Page 5, we add the following sentence:

“Photoluminescence (PL) of the pristine QDs was strongly quenched in the doped samples (Supplementary Fig. 3).”

3. Is it possible to observe those interesting behavior, namely slow hot electron relaxation in the solid state?

Response: We thank the reviewer for this suggestion. We have performed the measurements for n-doped QD films and the observations are similar to those of the solution samples. One issue, with the film samples, is that the optical scattering prohibits us from determining the nominal number of doped band-edge electrons in these samples based on the absorption spectra. Nonetheless, we can still semi-quantitatively control the number of doped electrons by controlling the amount of reducing reagent used.

Fig. R3a-c show the results for a relatively lightly-doped QD film. In this sample, by extracting the difference between the 1S bleach kinetics measured by 1P and 1S excitations, we can obtain the slow hot electron relaxation component (Fig. R3c). The lifetime is 12 ± 2 ps, which is similar to the spin blockade lifetime in the QD solution (9 ps).

Fig. R3d-f show the results for a relatively heavily-doped QD film. In this case, we also observe a 1S exciton absorption feature induced by 1P pump (Fig. R3e). This feature builds up in a few ps and decay within 100s of ps, similar to the behavior in the QD solution. The difference between the hot electron lifetimes in the film (~135 ps) and in the solution (~320 ps) indicates that there are additional hot electron relaxation channels in the solid state films.

Fig R3. (a) Absorption spectra of a lightly-doped QD film (black) and its corresponding neutralized film (red). (b) TA spectra of the lightly-doped QD film at indicated delays under 1P excitation. (c) TA kinetics of the lightly-doped QD film probed at the 1S bleach feature under 1S (blue squares) and 1P (red circles) pump conditions. Inset is the difference between the two kinetics (green triangles) and its fit (black solid line). (d) Absorption spectra of a heavily-doped QD film (black) and its corresponding neutralized film (red). (e) TA spectra of the heavily-doped QD film at indicated delays under 1P excitation. (f) TA kinetics of the heavily-doped QD film probed at the 1S absorption feature under 1P pump (blue squares). The black solid line is a fit to the decay kinetics.

4. How to evidence the population of surface states after photochemical doping?

Response: We thank the reviewer for this insightful comment. Direct evidence for electrons doped into surface states could be obtained by performing the electron titration experiment as done by Gamelin et al. in *J. Am. Chem. Soc.* 2013, 135, 18782-18785. They found that, for a doped CdSe QD sample with a nominal number of ~ 2 electrons per dot, up to 30 equiv of $[\text{FeCp}^*_2][\text{BArF}]$ ($[\text{FeCp}^*_2]^+ =$

decamethylferrocenium, $[\text{BArF}]^- = \text{tetrakis}[3,5\text{-bis}(\text{trifluoromethyl})\text{phenyl}]\text{borate}$ was required to neutralize the sample.

However, $[\text{FeCp}^*_2][\text{BArF}]$ is not commercially available and its synthesis is beyond our capability. One possibility is to contact Gamelin group for a collaboration, which, however, becomes almost impossible at this time of COVID-19 pandemic. We would like to note that the QDs used in this study were synthesized using a similar method as that used in Gamelin's study, meaning that the QDs should have similar surface properties. Therefore, we believe that the phenomenon reported there should also be applicable to our sample.

5. Please also make some comments on the charge extraction of hot electrons from colloidal QDs to metal oxides, which is critical for practical utilization in photon-electron conversion.

Response: We thank the reviewer for this suggestion. In our prior study (*Nat. Commun.* **2019**, *10*, 4532) on Cu-doped CdSe QDs which also displayed a long hot electron lifetime (~ 8.6 ps), we have studied QD-sensitized TiO_2 and ZrO_2 films. We found that on ZrO_2 films which served as a control, the long hot electron lifetime was still observed, indicating that incorporation into devices should not significantly alter the hot electron lifetime. On TiO_2 films, we also obtained evidence for hot electron transfer from Cu-doped CdSe QDs to TiO_2 .

The hot electron lifetime observed in heavily n-doped QDs was much longer than that in Cu-doped QDs. So we can expect that hot electron extraction from n-doped QDs to TiO_2 should be very efficient.

Revision: Per the reviewer's comments 3 and 5, we add the following contents on pages 16-17:

“Importantly, the phenomena of spin blockade and phonon bottleneck can also be observed in n-doped solid-state films of CdSe QDs (Supplementary Fig. 14), suggesting the possibility of utilizing the hot electrons in devices. In particular, QD-sensitized metal oxide (such as TiO_2) films offer a promising platform for hot electron extraction from QDs owing to the strong electronic coupling between QDs and metal oxides as well as the large density of electron-accepting states in the conduction band of metal oxides. For example, electron transfer from CdSe QDs to mesoporous SnO_2 films was reported to occur in a few picoseconds;⁵⁷ electron transfer from PbS and PbSe QDs to TiO_2 could be promoted to the femtosecond timescale by engineering the interfacial coupling.^{24,58} Our prior study also showed evidence for hot electron transfer from copper-doped CdSe QDs (hot electron lifetime ~ 8.6 ps) to TiO_2 . Such hot electron extraction should become even more efficient with the heavily n-doped QDs here featuring a hot electron lifetime as long as 300 ps.”

Reviewer #2 (Remarks to the Author):

In this study, Wang et.al. investigate ultrafast hot carrier dynamics in photochemically-doped CdSe quantum dots (QDs). They claim a spin-block effect for relatively-low doped QDs (~ 0.5 electrons in the 1Se state), which prolonged the 1Pe to 1Se hot carrier decay time from sub-ps to nearly 10 ps. For highly-doped QDs, they observe that some fraction of hot carriers can undergo very slow cooling with a decay lifetime exceeding ~ 300 ps. They explain their data by a phonon bottleneck effect for hot electrons following Auger recombination, which removes the photogenerated hole states. The paper deals with an important topic which could trigger interests from a broad range of audience. Data for highly doped QDs are fascinating (in figure 5 c-d) and new. However, I do not recommend the paper for Nature communications, because of (1) no clear evidence for spin-blockade effect for low-doped QDs based on their results, and (2) insufficient data analysis and discussion to support the photon bottleneck effect.

Response: We thank the reviewer very much for his/her positive comments on our work and for expressing his/her current concerns that should help us improve the paper. Below we provide our point-to-point responses and revisions to address these concerns.

Some of my major comments are the following:

(1) For the low-doped QDs, the only experimental observation that seems to be consistent with spin-blockade is the relatively prolonged hot carrier relaxation time. However, other effects could also lead to slow cooling by electron doping. For instance, if the photo-excited holes recombine with the electrons injected into defect states non-radiatively, the hot electrons can also be long-lived as the electron-hole energy transfer pathway is suppressed. The authors have claimed a critical role of defects for the highly-doped QDs, but completely neglect this for the low-doped ones.

Response: We thank the reviewer for this comment. The key difference between the heavily- and lightly-doped QDs is their Auger recombination lifetime, as fast as 1 ps in the former and 460 ps in the latter. As such, in the lightly-doped QDs, hot electron can relax down to the band edge via spin-flip and electron-to-hole energy transfer before Auger recombination.

Revision: On page 14, we add the following sentence to clarify this point:

“Note that in the n -doped-1 sample studied above, the Auger recombination process (460 ps) is too slow to compete with the spin-flip process, thus prohibiting the observation of the phonon bottleneck effect in lightly-doped QDs.”

Therefore, for a clear experimental demonstration of spin-blockade effect, some further discussions or even experimental pieces of evidence are required: (a) as the doped electrons are not spin-polarized, and photogenerated electron and hole are produced by a linearly-polarized pulse, 50% of doped QDs should exhibit the spin-blockade effect (if it is active). Can the author do further data analysis to check

this point? In principle, this can be done by applying e.g. Poisson statistics for photoexcitation. The normalized dynamics shown in fig. 4 (b) and (d) do not provide sufficient information on the fraction of charge carriers following the slow decay channel. In fact, by closely looking at the dynamics, the slow decay claimed by the authors is only a small portion of the dynamics. The majority of charge carriers, even following 1P excitation, fill the 1Se states in sub-ps.

Response: We thank the reviewer for this insightful comment. Indeed, under our experimental conditions, there is only a small portion of hot electrons that displayed slow relaxation. But this small portion is consistent with the estimation using the population analysis accounting for statistics of electron doping. Specifically, the average number of electrons per QD is 0.5 for this sample. Assuming a Poisson distribution of the doped electrons, ~60% of the QDs were undoped, ~30% of the QDs were doped with one electron and ~10% were doped with ≥ 2 electrons. QDs doped with ≥ 2 electrons would not display exciton bleach on the TA spectra because the band edge exciton absorption was already fully blocked. Among the QDs doped with one electron, only half of them would experience a spin blockade due to random distribution of the spin directions. Thus, the portion of the TA signal that would correspond to the spin blockade effect is only $(1/3)*(1/2) = 1/6$ of the total signal amplitude. According to Fig. 4d, the total amplitude under 1P pump is about -8 mOD, and therefore the spin blockade should correspond to -1.33 mOD. This is roughly consistent with the amplitude of the slow formation component (~1.2 mOD) observed in Fig. 4d.

We also analyzed the statistics for photon absorption, as suggested by the reviewer, by performing a power-dependent TA signal saturation experiment. In this experiment, we measured a neutral QD sample which has the same absorption as the n-doped-1 sample at the pump wavelength (547 nm). The underlying principles are that the photon absorption statistics at a photon energy much higher than the bandgap is a Poissonian and that after multiexciton Auger recombination the TA signal amplitude is simply proportional to the fraction of photoexcited QDs in the ensemble $(1 - e^{-\langle N \rangle})$, $\langle N \rangle$ the average number of photons absorbed per QD). $\langle N \rangle$ is proportional to the absorption cross section and excitation fluence. Thus, by fitting the TA signal amplitude at 600 ps (Auger finished) as a function of the excitation fluence, we can determine $\langle N \rangle$ at any specific pump fluence; see Fig. R4. The experiment results shown in Fig. 4c and 4d correspond to a pump fluence of $11.2 \mu\text{J}/\text{cm}^2$, and hence $\langle N \rangle$ of ~0.017. At this $\langle N \rangle$, 98.3% of QDs were unexcited, 1.7% of QDs were excited with one exciton and the QDs excited with 2 or more excitons were negligible. This point guarantees that our analysis in the above paragraph does not need to include multiexciton effects.

Fig. R4. Scaled TA signal amplitude at 600 ps, after multiexciton Auger recombination has finished, as a function of the excitation fluence of the 547 nm pump (blue squares). The saturation curve is fitted to a Poissonian model for photon absorption (black solid line), from which we can determine $\langle N \rangle$, the average number of photons absorbed per QD, at any specific pump fluence (red dashed line).

(b) to fundamentally prove spin-blockade, further experiments based on spin-polarized electrons injected into 1S_e states are encouraged. By further changing the circular polarization of the pump, one should see a drastic change in hot carrier lifetimes. While the comment (b) seems challenging, further discussions for (a) is mandatory.

Response: We thank the reviewer for this suggestion. While excitation using circularly-polarized light can indeed inject spin-polarized electrons, it does not help with the spin blockade in our system. The reason is that the doped electrons are not spin-polarized. Due to a random distribution of the spins of the doped electrons, a spin-polarized 1P electron will still have 50% chance to have the same spin as the doped electron and the other 50% to have the opposite spin.

Revision: Per the reviewer’s suggestion, on page 9, we add the following contents: “Specifically, the average number of doped electrons per QD is ~ 0.5 for the n -doped-1 sample. Assuming a Poisson distribution of the doped electrons,³⁸ $\sim 60\%$ of the QDs were undoped, $\sim 30\%$ were doped with one electron, and $\sim 10\%$ were doped with ≥ 2 electrons. QDs doped with ≥ 2 electrons would not display 1S exciton bleach on the TA spectra because the absorption was already fully blocked. Among the QDs doped with one electron, statistically only half of the photoinjected 1P_e electrons would have the same spin as the doped 1S_e electron due to random distribution of the spin directions. Note that even if we create spin-polarized 1P_e electrons using circularly-polarized pump pulses, this statement still holds because there is no control over the spin directions of pre-doped 1S_e electrons. Thus, the portion of the 1S bleach amplitude that would correspond to the spin blockade effect is only $1/6$ ($= 1/3 \times 1/2$) of the total 1S bleach amplitude. According to Fig. 4d, the total 1S bleach amplitude

under 1P pump is ~ 8 mOD, and therefore the spin blockade signal can be estimated as ~ 1.3 mOD, which is consistent with the amplitude of the slow formation component (~ 1.2 mOD) observed in Fig. 4d.

The agreement between the estimated and measured signal amplitudes above provides strong evidence that the slow $1P_e$ relaxation component of 9 ps can be ascribed to a Pauli spin blockade induced by the preexisting $1S_e$ electron (Fig. 1b left)."

Finally, the spin-blockade concept is not new and has been proposed and demonstrated previously (J. Phys. Chem. Lett. 2019, 10, 2341–2348);

Response: We thank the reviewer for pointing out this issue. But we have never claimed that the spin-blockade is a new concept here. We cited and discussed the referred papers in our prior submission: "*Interestingly, a recent study of pump-power dependent TA kinetics of CdSe/CdS core/shell QDs also implies a spin blockade for $1P_e$ electron relaxation with a time constant of 25 ps,⁵³ this is longer than the time constant in our sample likely because the shell can slow down spin flip.*"

The novelty of our study is that we identified n-doped QDs as a platform to realize the spin-blockade effect. In the JPCL paper, the blockade was created by exciting the QDs with excitons, which are short-lived species (ns timescale). In contrast, our n-doped QDs are free-standing samples that are stable in the steady state.

Revision: On page 10, we add the following sentence:

"It is worth mentioning that, in that study the spin-blockade was realized by transiently exciting the QDs with excitons, whereas herein we observe the effect in free-standing n-doped QDs that are stable in the steady state."

(2) For highly-doped QDs, while a slow decay in the dynamics over 300 ps seems clear from the dynamics, a critical point that is relevant for application is what percentage of charge carriers follow this path? Some quantitative analysis is necessary (e.g. by applying Poisson statistics for the carrier population following excitation) based on the signal strength (not just the normalized dynamics).

Response: We thank the reviewer for this useful suggestion. The signal amplitude analysis can be performed using the method described above. To begin with, the average number of photons per QDs is still $\ll 1$, excluding complications from multiexciton effects. Next, the average number of electrons doped per QDs in the heavily-doped sample is ~ 1.3 . Using Poisson statistics, the percentages of QDs doped with 0, 1 and ≥ 2 electrons are 27.2%, 35.4% and 37.3%, respectively.

i) In QDs doped with ≥ 2 electrons, the photoinjected $1P_e$ electron cannot relax until a doped $1S_e$ electron is consumed by Auger recombination. Once Auger occurred, a valence band hole is also consumed, shutting down the electron-to-hole energy transfer pathway. Therefore, this part of QDs should all display long-lived hot electrons owing to the phonon bottleneck effect.

ii) For QDs doped with one electron, half of them should display rapid (< 1 ps) hot electron relaxation (no spin blockade effect), whereas the other half should experience the spin blockade. The lifetime for the blockade, according to the study on

lightly-doped QDs, is ~ 9 ps. This process competes with the Auger recombination and the competition determines the partition between QDs that relax via spin-flip and those showing the phonon bottleneck. The Auger recombination has two exponential components, and, to simplify the estimation, we use the amplitude-averaged time constant of 7.6 ps. Thus, the ratio between QDs that relax via spin-flip and those showing the phonon bottleneck is 45.8%:54.2%.

iii) For undoped QDs, their 1P_e hot electrons rapidly relax via electron-to-hole energy transfer.

Therefore, the total percentage of the QDs that displays the phonon bottleneck effect is estimated as: $37.3\% + 35.4\% \times 54.2\% / 2 = 46.9\%$. In the other QDs, the rapid relaxation through electron-to-hole energy transfer and/or spin-flip should result in fast decay of the 1P bleach within ~ 50 ps. According to the curve in Fig. 5d, the percentage of the long-lived components of the 1P kinetics is $\sim 44.8\%$ (3 mOD/6.7 mOD). This is in reasonable agreement with the estimated ratio.

(2a) There is a large portion of fast decay for the 1P bleach, which the author attributed to “both Auger recombination and spin-flip relaxation”. Why is that? Why did these effects not matter for the low-doped QDs?

Response: This point has been explained above by considering the relaxation dynamics in QDs doped with different numbers of electrons.

Revision: Per the reviewer’s suggestions, on pages 13-14, we add the following contents:

“There are fast decay components for the 1P bleach (Fig. 5d), resulting from inhomogeneous distribution of the predoped electrons in the ensemble sample. The average number of electrons doped per QD in the heavily-doped sample is ~ 1.3 . Using Poisson statistics, the percentages of QDs doped with 0, 1 and ≥ 2 electrons are 27.2%, 35.4% and 37.3%, respectively. In QDs doped with ≥ 2 electrons, the photoinjected 1P_e electron cannot relax until a doped 1S_e electron is consumed by Auger recombination; thus all these QDs should display long-lived hot electrons. For QDs doped with one electron, half of them should display rapid 1P_e relaxation (no spin blockade), whereas the other half should experience the spin blockade. The spin-flip process (time constant ~ 9 ps) competes with the Auger recombination (amplitude-averaged time constant ~ 7.6 ps) and this competition determines the partition between QDs that relax via spin-flip and those showing the phonon bottleneck (45.8%:54.2%). In the undoped QDs, all the 1P_e hot electrons rapidly relax via electron-to-hole energy transfer. Thus, the overall percentage of QDs displaying long-lived hot electrons is estimated as: $37.3\% + 35.4\% \times 54.2\% / 2 = 46.9\%$. According to Fig. 5d, the percentage of the long-lived components of the 1P kinetics is $\sim 44.8\%$ (3.0 mOD/6.7 mOD), which is in good agreement with the estimated ratio. Note that in the *n*-doped-2 sample studied above, the Auger recombination process (460 ps) is too slow to compete with the spin-flip process, thus prohibiting the observation of the phonon bottleneck effect in lightly-doped QDs.”

Furthermore, I found some claims for discussion are not well supported or inconsistent for the main conclusion of “phonon bottleneck” effect:

(2b) For the scheme in the figure 5b, what happened to the electrons in the defect following the Auger process following 1S excitation? In the scheme, this electron just disappeared and the authors did not comment on this. The electron can be excited to high energy states following Auger excitation, and should stay hot following the “phonon bottleneck” scenario proposed in the paper. However, the dynamics show only an ultrafast decay.

For the scheme in the figure 5d, before photoexcitation there is 1 electron in 1S state, and arbitrarily 4 electrons in the defects. In the end, following Auger and relaxation, the system did not decay back to the initial states with only 3 electrons in the defects. Why is that? In any case, following what has been shown in both schemes in figure 5 b and d, one should expect long-lived hot carriers in both cases. But this is not in line with experimental results.

Response: We thank the reviewer very much for this insightful comment. As pointed out by the reviewer, we did not observe slow growth again of the 1S bleach after Auger recombination. It is thus interesting to examine the fate of the electron excited in an Auger process.

One possibility is that the electron is transiently ejected outside the QD after accepting the energy released by electron-hole recombination (i.e., Auger ionization) and eventually returns to the surface states. The other possibility is that the electron is excited to a very high level either in the conduction band or in the surface states. In this case, if the electron relaxes back to the band edge, we should expect slow growth again of the 1S bleach as a manifestation of the phonon bottleneck effect, which, however, is not observed in our experiment (Fig. 5b). Therefore, we suspect that the highly-excited electron also returns to the intra-gap surface states.

We thank the reviewer for reminding us that the schemes in Fig. 5b and 5d lacked the details for the relaxation of the excited electron. We have now revised these schemes to clarify it.

Revision: On page 13, we add the following contents to discuss the Auger-excited electron:

“It is interesting to examine the fate of the surface-doped electron that is excited in an Auger recombination event (Fig. 5b scheme). One possibility is that it is transiently ejected outside the QD after accepting the energy released by electron-hole recombination (i.e., Auger ionization) and eventually returns to the surface states. The other possibility is that the electron is excited to a very high level either in the conduction band or in the surface states. In this case, if the electron relaxes back to the band edge, we should expect slow growth again of the 1S bleach as a manifestation of the phonon bottleneck effect, which, however, is not observed in our experiment (Fig. 5b). Therefore, we suspect that the highly-excited electron also returns to the intra-gap surface states. The “invisibility” of the electrons in the surface states prohibits conclusive statements on the relaxation pathways of the excited electron.”

(2c) Auger rates throughout the paper: for the low-doped QDs, the authors reported an Auger lifetime of 460 ps. This value is way higher than what has been reported previously (in the range of tens ps for the same diameter, see in figure 3 in the paper by Klimov et. al. Science 2000, 1011-1013). Why is that? Can the dynamics be interpreted differently? Can we normalize the dynamics to the early time (rather than the later time scale)? By doing so, we will see a faster decay for doped QDs in the range of 100-1000 ps, which may be attributed to e.g. trapping.

Response: We thank the reviewer for this comment. We would like to clarify that when talking about the “Auger lifetime” of a QD, it is necessary to note which type of excited states we are referring to. The Auger lifetimes in Klimov’s Science paper are lifetimes of neutral multiexciton states, such as biexcitons, triexcitons... In contrast, the Auger lifetimes we talked about in our paper are those of charged single-excitons.

There have been extensively studies on the relationships between the Auger lifetimes of biexcitons and positive and negative trions (see, e.g., *Annu. Rev. Cond. Matt. Phys.* **2014**, 5, 285-316. According to a simple statistical argument, the Auger rate of a biexciton is two times the sum of the rates of a positive trion and a negative trion. In addition, for II-VI group QDs, Auger of a negative trion is slower than a positive trion. For these reasons, the Auger lifetime of a negative trion can be much longer than that of a biexciton. This has been experimentally demonstrated in a prior paper by Gamelin group (*Nano Lett.* **2014**, 14, 353-358).

In fact, the lifetime we obtained (460 ps) is not too long but is shorter than the trion lifetime of QDs of similar size (740 ps) in the Gamelin paper. This has been discussed in our prior submission: “*In principle, because n-doped-1 has a low doping level of 0.5 $1S_e$ electron per QD, the measured Auger lifetime should be that of a negative trion (X^-). However, the lifetime of X^- previously reported for 5.5-nm CdSe QDs coated with a thin ZnS shell is 740 ps.⁴⁹ The inconsistency lies in the ZnS shell that effectively alleviates trap states on CdSe surfaces. In our core-only QDs, surface states, especially electron-trapping states, can be filled by electrons during photochemical doping.⁴⁴ These trap-state electrons could accelerate the rate of Auger recombination as this rate scales up with the number of carriers⁵².*”

As for the normalization procedure, the reason why we normalize the kinetics at the later time is that there are undoped QDs in the ensemble. These undoped QDs should display long-lived single exciton trapping/recombination only similar to those of a neutral sample. Therefore, by normalizing the kinetics of a doped sample and a neutral sample at the later time, we can remove the contribution from the undoped QDs in the doped sample such as to obtain the recombination kinetics in the doped QDs. This is also explained already in our prior submission: “*Because the doped sample also contains undoped QDs, we can normalize the 1S bleach kinetics of neutral and doped samples to a long-lived tail and then subtract the former from the latter to isolate the kinetics of pure doped QDs. Fitting the kinetics reveals an Auger-dominated lifetime constant of 460 ± 30 ps (Fig. 4b inset).*”

Revision: On page 8, we add the following sentence:

“note the difference between the Auger lifetimes of X^- and neutral biexcitons⁵¹.”

In contrast, for the highly-doped QDs, the Auger lifetime is as short as 1 ps. How and why the Auger lifetime can be tuned more than two orders of magnitude by adding ~ 0.8 more electrons/QDs (by comparing low (0.5 1Se electron) and high doping (1.3 1Se electron)? Based on the paper from Klimov (Science 2000, 1011-1013), compared to the first order Auger process, the rates for high order Auger processes should increase, but not as dramatic as reported here. Although the authors claimed that “there should be many more electrons doped into the trap states that strongly accelerate Auger recombination”, this should also apply to the low-doped case. The defects must be filled before they can host electrons in the 1Se states (0.5 for low-doped case). This is a critical point, as a few discussions made by the authors are based on such assignments in dynamics: e.g. “Note that there is a fast decay component for the 1P bleach that has complicated origins from both Auger recombination and spin-flip relaxation.”

Response: We thank the reviewer for this very insightful comment. Indeed, we are also surprised by that the Auger recombination in the heavily-doped sample is as fast as 1 ps, whereas that in the lightly-doped sample is 460 ps. As pointed out by the reviewer, surface-state electrons should exist in both samples and, in principle, the numbers of these electrons should be similar because the trap states have to be filled before the electrons can be doped into the conduction band edge.

One possibility to rationalize the contradiction is to consider the change of density of surface states induced by the doping chemistry. The doping reagent we used, LiEt₃BH, is a strong reductant, and it could actually perturb the binding between QDs and surface ligands. In our experiment, we have used ~ 2 and 9 drops of the LiEt₃BH/tetrahydrofuran/toluene solution for the lightly and heavily-doped samples, respectively. Therefore, it is likely that we have introduced many more trap states in the heavily-doped sample compared to the lightly-doped one. As a result, the number of electrons doped into the trap states in the heavily-doped sample is also much larger compared to the lightly-doped one.

Admittedly, we don't have direct evidence for the speculation above. But the Auger lifetimes we determined are highly reproducible, and the phenomenon that Auger recombination in heavily-doped samples are much faster than that in light-doped ones is also highly reproducible among many different QD samples (Supplementary Figs. 8 to 13).

Revision: On page 11, we add the following contents:

“It is surprising that the Auger recombination in n -doped-2 is so much faster than that in n -doped-1. In principle, all the trap states have to be filled before the electrons can be doped into the conduction band edge, and hence, the numbers of trap-state electrons should be similar in both samples. Further, the numbers of band edge electrons in both samples are not strongly different either (0.5 versus 1.3), contradicting the orders-of-magnitude difference in their Auger lifetimes. One possibility is that the densities of trap states are not constant but rather vary with the doping conditions. The strong reducing reagent, LiEt₃BH, could perturb the binding

between QDs and surface ligands, thus introducing additional trap states to the QDs. Because the amount of LiEt₃BH applied to *n*-doped-2 is much larger than *n*-doped-1 (see Methods), we suspect that there are many more surface-state electrons in the former than the latter.”

(2d) In the introduction, the authors claimed that “Alternatively, a doped 1Se electron can be directly excited to the 1Pe level using a mid-infrared photon; the resulting 1Pe electron is fully decoupled from any electron or hole and it should be very long-lived because of a phonon bottleneck (Fig. 1b, right).” While it is a bit misleading (as the authors did not follow such an intra-band excitation method later in the paper), I find this statement fascinating: it provides a clear route for demonstrating the phonon bottleneck effect. However, in reality, the author follows an “inter-band” excitation scheme, which complicates the data analysis. To fully support the conclusion of the phonon bottleneck effect, I would recommend the author to conduct such studies. It should not be that difficult, as one needs just to excite the doped QDs below their optical bandgap.

Response: We thank the reviewer for this very useful suggestion. We have now performed this experiment and the results are summarized in our response to reviewer 1’s comment 1.

Other comments to the paper:

(3) For low-doped QDs, in the data analysis part, subtraction of two dynamics following 1S and 1P excitations have been applied. This implies a common decay pathway shared by 1P and 1S excitation. What is the origin of the common dynamics?

Response: We thank the reviewer for this question. The common origin is the recombination dynamics of single-excitons (including Auger recombination of charged single excitons in the doped QDs and recombination of neutral single-excitons in undoped QDs in the *n*-doped-1 ensemble sample). The difference between 1S and 1P excitation conditions only lies in the growth kinetics of the 1S bleach feature; a subtraction between them can thus reflect the growth kinetics.

Revision: On pages 8-9, we revised the following sentence:

“They share the same decaying kinetics, with the decay reflecting Auger recombination of charged single-excitons in the doped QDs and recombination of neutral single-excitons in the undoped QDs in the *n*-doped-1 ensemble sample, but in the case of 1P excitation there is a slow growth process persisting until ~30 ps.”

(4) On page 9, the authors stated that “This decay annihilates a 1Se electron, deviating from our initial expectation that the 1Pe electron would be long-lived due to full occupation of the 1Se level.” This statement is very confusing. Here you have excited the system only to the 1Se states. Why will this lead to long-lived 1Pe (because you did not excite it)?

Response: We thank the reviewer for pointing out this confusing sentence. What we actually mean is that even if we can fully block the 1Se level using two electrons, one

of the electrons will rapidly decay due to Auger recombination, opening a vacancy in the 1Se level for the relaxation of a 1Pe electron. We agree with the reviewer that it is kind of confusing if we put it in its current position. Therefore, we have removed this sentence.

(5) In the introduction, the authors claimed that “Hot electrons are unthermalized electrons that carry large kinetic energies.” This is not correct, as hot electrons can be thermalized or non-thermalized. In fact, the connotation ‘hot’ implies there is a temperature associated with the electron distribution. Some rephrasing is needed.

Response: We thank the reviewer for pointing out this issue. We fully agree with the reviewer’s statement.

Revision: We have rephrased the sentence to:

“Hot electrons carry large kinetic energies compared to band edge electrons.”

(6) On page 10, the authors mentioned that “The build-up of this signal is on a few ps timescale (Fig. 5d), which is similar to the Auger recombination time shown in Fig. 5b.” What is exactly the build-up time? Why is it faster than the Auger time (~ 1ps) for 1S pump? Given that the same Auger processes occur following both 1S and 1P excitations sketched by the authors, I would expect the Auger time is the same here for both 1S and 1P excitations.

Response: We thank the reviewer for this comment. Indeed, the build-up time of the 1S absorption in the case 1P pump (Fig. 5d) is consistent with the fast component of the Auger recombination time measured under 1S pump (the 1 ps process displayed in Fig. 5b). We called them similar in our prior submission because there is also a slow component of 31 ps for Auger recombination in Fig. 5b, which is not reflected in the build-up time of the 1S absorption in the case 1P pump because it is convoluted with some fast hot electron relaxation components explained in our response to the reviewer’s comment (2). As shown in Fig. R5, if we scale and downshift the 1S build-up kinetics to selectively compare its fast component with the Auger recombination process, we can find that they are indeed the same.

Fig. R5. Comparison between the 1S bleach decay kinetics measured under 1S pump condition and the 1S absorption build-up kinetics measured under 1P pump condition. The latter has been rescaled and downshifted in order to compare with the fast decay component only of the Auger process.

Revision: We have revised the related contents to:

“The build-up time of this signal (Fig. 5d) is consistent with the time constant of the major component of the Auger recombination process measured in Fig. 5b. The minor, slow component of Auger recombination is not reflected on the build-up kinetics in Fig. 5d because of a convolution between this slow formation and some fast decay components that will be explained later.”

REVIEWER COMMENTS

Reviewer #1 (Remarks to the Author):

The authors have addressed all my questions.
I suggest it can be published.

Reviewer #2 (Remarks to the Author):

In the revised manuscript entitled "Spin blockade and phonon bottleneck for hot electron relaxation observed in n-doped colloidal quantum dots", Wang and his colleagues have made a great effort to provide further experimental evidence on phonon bottleneck effect in heavily doped QDs, and deepen some discussions following the comments of the two referees. These have increased the quality as well as the impact of the paper drastically. I am particularly satisfied with the new data present in figure S6: the author did an "intraband-band" pump (by a mid-IR pulse), and 1S interband probe study, which provides direct evidence for slow intraband-band relaxation over 100s ps time scale. This is also consistent with their previous results following 1P excitation -1P or 1S probe studies. From this perspective, I support the paper to be published in Nature communications. However, before the paper can be accepted, I strongly urge the authors to address a few follow-up comments.

(1) Figure S6 is a critical result, which should be included in the main paper, e.g. into figure 5. In fact, this intraband excitation result provides a much more direct evidence than the "interband" excitation studies in the current figure 5.

(2) The relaxation scenarios presented in Figures 1 and 5 are still a bit confusing. For instance, in figure 1 for the spin-blockade case: from the right to the left panel, there is a sudden increase of electron number from 2 to 3. What is shown in the right panel is more like an effect of "Coulomb blockade" due to state filling. For the phonon bottleneck effect panel, how it is shown now, is like direct doping to 1P state in QDs. I think that it would be very useful to show that this is achieved by an intraband excitation to QDs with initial doping to the 1S state (by one or two panels). For figure 5: in the upper right panel, the initial doping of QDs for 1S excitation consists of 1 electron in the 1S state, while this number increase to 2 for the bottom right panel following 1P excitation. It did take me a while to fully grasp it: that there are distributions in the number of electrons in an individual QDs, and that the related physical processes (Auger process in figure 5b and phonon bottleneck effect in figure 5d) have different sensitivity to the doping level or number of electrons/QD. So I would suggest that either authors clarify it much earlier part into the paper to better guide the reader, or present the schematics with the same initial conditions: e.g. with one electron doped in the 1S state. This won't change the scenario which the author would like to propose, yet keep everything relatively simple.

(3) The size-dependent data in figure 6a-b do not add much value, as both results are size-independent for the sizes used here. I would suggest to put them into SI.

(4) Regarding the estimation on the relative spectral weight of the phonon bottleneck effect from line 271-289. I did not follow the discussion on the 1 electron doping scenario: "For QDs doped with one electron, half of them should display rapid 1Pe relaxation (no spin blockade), whereas the other half should experience the spin blockade. The spin-flip process (time constant ~ 9 ps) competes with the Auger recombination (amplitude-averaged time constant ~ 7.6 ps) and this competition determines the partition between QDs that relax via spin-flip and those showing the phonon bottleneck (45.8%:54.2%)." Why is the spin blockade relevant here? The Auger process is shown to be ultrafast ~ 1 ps from figure 5b for the heavily doped QDs. This will bring away a 1S, or a 1P electron and the

only hole in the QDs. This is sufficient to set the system (with 50% chance) to "intraband excitation" configuration, in which an electron is excited to the 1P state, following by a phonon bottleneck effect. In addition, where is this amplitude-averaged time constant ~ 7.6 ps (for Auger) coming from?

(5) Some of replies in the rebuttal are still not fully satisfying: (a) for my first question regarding "For the low-doped QDs, the only experimental observation that seems to be consistent with spin-blockade is the relatively prolonged hot carrier relaxation time. However, other effects could also lead to slow cooling by electron doping. For instance, if the photo-excited holes recombine with the electrons injected into defect states non-radiatively, the hot electrons can also be long-lived as the electron-hole energy transfer pathway is suppressed. The authors have claimed a critical role of defects for the highly-doped QDs, but completely neglect this for the low-doped ones." The authors replied by claiming the Auger process is slow so that can not compete with the spin-flip, but this is not what I asked; (b) The same goes to the reply to the question 1 from the other referee: the author did not reply to the question;

(5) It should be „possibility“ rather than „possibly“ in the sentence: " One possibly is that the densities of trap states are not constant but rather vary XXX", starting from line 229

Reviewer comments in black, our responses in red and revisions in blue

REVIEWER COMMENTS

Reviewer #1 (Remarks to the Author):

The authors have addressed all my questions.

I suggest it can be published.

Response: We thank the reviewer very much for recommending our paper for publication.

Reviewer #2 (Remarks to the Author):

In the revised manuscript entitled "Spin blockade and phonon bottleneck for hot electron relaxation observed in n-doped colloidal quantum dots", Wang and his colleagues have made a great effort to provide further experimental evidence on phonon bottleneck effect in heavily doped QDs, and deepen some discussions following the comments of the two referees. These have increased the quality as well as the impact of the paper drastically. I am particularly satisfied with the new data present in figure S6: the author did an "intraband-band" pump (by a mid-IR pulse), and 1S interband probe study, which provides direct evidence for slow intraband-band relaxation over 100s ps time scale. This is also consistent with their previous results following 1P excitation -1P or 1S probe studies. From this perspective, I support the paper to be published in Nature communications. However, before the paper can be accepted, I strongly urge the authors to address a few follow-up comments.

Response: We thank the reviewer very much for supporting our revised paper for publication. Below we provide our point-to-point responses and revisions to address his/her remaining concerns.

(1) Figure S6 is a critical result, which should be included in the main paper, e.g. into figure 5. In fact, this intraband excitation result provides a much more direct evidence than the "interband" excitation studies in the current figure 5.

Response: We thank the reviewer for this suggestion. We have now included Fig. S6 into Fig. 5.

Revision: The new Fig. 5 now looks like:

(2) The relaxation scenarios presented in Figures 1 and 5 are still a bit confusing. For instance, in figure 1 for the spin-blockade case: from the right to the left panel, there is a sudden increase of electron number from 2 to 3. What is shown in the right panel is more like an effect of “Coulomb blockade“ due to state filling. For the phonon bottleneck effect panel, how it is shown now, is like direct doping to 1P state in QDs. I think that it would be very useful to show that this is achieved by an intraband excitation to QDs with initial doping to the 1S state (by one or two panels).

Response: We thank the reviewer for these suggestions. We have now revised Fig. 1b to clarify these issues.

Revision: The new Fig. 1 now looks like:

For figure 5: in the upper right panel, the initial doping of QDs for 1S excitation consists of 1 electron in the 1S state, while this number increase to 2 for the bottom right panel following 1P excitation. It did take me a while to fully grasp it: that there are distributions in the number of electrons in an individual QDs, and that the related physical processes (Auger process in figure 5b and phonon bottleneck effect in figure 5d) have different sensitivity to the doing level or number of electrons/QD. So I would suggest that either authors clarify it much earlier part into the paper to better guide the reader, or present the schematics with the same initial conditions: e.g. with one electron doped in the 1S state. This won't change the scenario which the author would like to propose, yet keep everything relatively simple.

Response: We thank the reviewer for these suggestions. We have now revised Fig. 5 caption as well as some main text contents to clarify these issues.

Revision: In captions of Fig. 5b and 5d, we add the following explanations, respectively:

“Note that only QDs doped with one electron are shown because QDs doped with two or more electrons cannot absorb the 1S resonant pump photon.” (Fig. 5b)

“Only QDs with two doped electrons are shown in this scheme. In QDs doped with one electron, there is a competition between Auger recombination and spin-flip hot electron relaxation; see main text for details.” (Fig. 5d)

(3) The size-dependent data in figure 6a-b do not add much value, as both results are size-independent for the sizes used here. I would suggest to put them into SI.

Response: We thank the reviewer for this suggestion. But we feel keeping the different-size data here is also appropriate. At least they demonstrate how robust our observations are. We hope the reviewer can agree with it.

(4) Regarding the estimation on the relative spectral weight of the phonon bottleneck

effect from line 271-289. I did not follow the discussion on the 1 electron doping scenario: “For QDs doped with one electron, half of them should display rapid 1Pe relaxation (no spin blockade), whereas the other half should experience the spin blockade. The spin-flip process (time constant ~ 9 ps) competes with the Auger recombination (amplitude-averaged time constant ~ 7.6 ps) and this competition determines the partition between QDs that relax via spin-flip and those showing the phonon bottleneck (45.8%:54.2%).“ Why is the spin blockade relevant here? The Auger process is shown to be ultrafast ~ 1 ps from figure 5b for the heavily doped QDs. This will bring away a 1S, or a 1P electron and the only hole in the QDs. This is sufficient to set the system (with 50% chance) to "intraband excitation" configuration, in which an electron is excited to the 1P state, following by a phonon bottleneck effect. In addition, where is this amplitude-averaged time constant ~ 7.6 ps (for Auger) coming from?

Response: We thank the reviewer for this comment. We would like to note that the Auger recombination process in Fig. 5b is clearly two-exponential instead of single-exponential. There is a slow component of 31 ps in addition to the 1 ps process. If there is only the 1 ps process, then as suggested by the reviewer, spin-flip cannot compete with Auger, and this would “set the system (with 50% chance) to "intraband excitation" configuration”. But if we consider the average Auger lifetime, the branching ratio should roughly follow the number we estimated.

Revision: On page 11, we have revised the following sentence:

“The decay is not single-exponential and can be fitted to a two-exponential decay function with a major component (78%) of 1 ps and a minor one (22%) of 31 ps; the amplitude-averaged lifetime is 7.6 ps (Fig. 5b).”

(5) Some of replies in the rebuttal are still not fully satisfying: (a) for my first question regarding “For the low-doped QDs, the only experimental observation that seems to be consistent with spin-blockade is the relatively prolonged hot carrier relaxation time. However, other effects could also lead to slow cooling by electron doping. For instance, if the photo-excited holes recombine with the electrons injected into defect states non-radiatively, the hot electrons can also be long-lived as the electron-hole energy transfer pathway is suppressed. The authors have claimed a critical role of defects for the highly-doped QDs, but completely neglect this for the low-doped ones.“ The authors replied by claiming the Auger process is slow so that can not compete with the spin-flip, but this is not what I asked;

Response: We feel sorry if our response has not fully addressed the reviewer’s question. If I understand it correctly now, the reviewer is proposing a scenario of ultrafast nonradiative recombination between the photoexcited hole and pre-doped defect state electron (not Auger) that can eliminate the hole and hence slow down electron cooling.

First, although such a recombination channel might exist, it should be relatively slow because of a weak wavefunction overlap between the photogenerated valence band hole and the pre-doped trap-state electron. We don’t think it can be as fast as the

sub-ps electron-to-hole energy transfer process. In fact, even in the heavily-doped sample, we don't see such a fast decay channel (the fast decay is Auger recombination rather than nonradiative electron-hole recombination). Secondly, we have performed a detailed population analysis on Page 9, per the reviewer's prior suggestion, and the analysis shows that the experimental amplitude ratio of the slow relaxation component is consistent with the one estimated using a spin blockade model. Last but not the least, if such a process indeed annihilated the hole, we would expect a phonon bottleneck effect resulting in hot electrons with lifetime of 100s of ps, rather than the 9 ps observed here. For these reasons, we hope with reviewer can agree with our proposed picture of spin blockade in the lightly-doped sample.

Revision: From page 10 to 11, we have added the following sentences:

“Although the population analysis above as well as comparison to literature strongly supports the assignment of the 9 ps slow relaxation to a Pauli spin blockade, an alternative possibility is that the nonradiative recombination between the photoexcited hole and pre-doped trap-state electron occurred on a sub-ps timescale, thus eliminating the hole before the electron-to-hole energy transfer. This would lead to a situation similar to the one depicted in the rightmost panel of Fig. 1b, i.e., a long-lived $1P_e$ electron enabled by a phonon bottleneck. However, as we will present later, this type of $1P_e$ electron will have a lifetime of 100s of ps.”

(b) The same goes to the reply to the question 1 from the other referee: the author did not reply to the question;

Response: We thank the reviewer for this comment. Admittedly, Reviewer 1's first question regarding the role of trap states in practical utilization of n-doped QDs is difficult to fully address; in fact, trap states are a general issue for the device applications of core-only QDs. Although Reviewer 1 has already been satisfied with our prior response, we make the following revision to further clarify this issue.

Revision: From page 14 to 15, we have revised the following sentences:

“Although the surface-state electrons and associated ultrafast Auger recombination have enabled the observation of a phonon bottleneck, they could also complicate the real application of *n*-doped QDs. Fortunately, the surface-state electrons are not necessarily required. As depicted in Fig. 1b (right), the phonon bottleneck can be directly realized using a mid-IR-pump, visible-probe TA experiment...”

(5) It should be „possibility“ rather than „possibly“ in the sentence: “ One possibly is that the densities of trap states are not constant but rather vary XXX“, starting from line 229.

Response: We have now fixed this typo.

REVIEWERS' COMMENTS

Reviewer #2 (Remarks to the Author):

The authors have addressed all my concerns, for which I'm grateful. The manuscript can be published, as far as I'm concerned.